# Affinity Group Experiences in Pharmacy Education: A Pilot Study

**DOI:** 10.3390/pharmacy13030070

**Published:** 2025-05-20

**Authors:** Elise Moore, Caroline Gaither, Olihe Okoro, Laura Palombi, L’Aurelle A. Johnson

**Affiliations:** 1College of Pharmacy, University of Minnesota—Minneapolis, 308 SE Harvard St, Minneapolis, MN 55455, USA; 2Department of Pharmaceutical Care and Health Systems, College of Pharmacy, University of Minnesota—Minneapolis, 308 SE Harvard St, Minneapolis, MN 55455, USA; cgaither@umn.edu; 3Department of Pharmacy Practice and Pharmaceutical Sciences, College of Pharmacy, University of Minnesota-Duluth, 1110 Kirby Dr, Duluth, MN 55812, USA; ookoro@d.umn.edu; 4Essentia Institute forRural Health, Essentia Health-Duluth, Essentia Health, 502 E 2nd St, Duluth, MN 55805, USA; 5Department of Experimental and Clinical Pharmacology, College of Pharmacy, University of Minnesota—Minneapolis, 308 SE Harvard St, Minneapolis, MN 55455, USA; joh02745@umn.edu

**Keywords:** affinity groups, underrepresented, sense of belonging, wellbeing

## Abstract

This study aims to examine the current wellbeing among pharmacy students in underrepresented groups (URGs) and investigate the impact on having access to affinity groups. A survey was distributed to students in April and May 2023, at a Midwestern College of Pharmacy, centering around diversity, equity, inclusion, and accessibility (DEIA) efforts and their impact on student wellbeing and experiences with the college-supported affinity groups. Student responses were analyzed using descriptive statistics. Sixty-five (75%) of the eighty-six students who completed the survey identified with at least one URG. First-generation students (*n* = 36), underrepresented racial/ethnic groups (*n* = 24), and LGBTQ+ (*n* = 13) were the three most prominent groups. Twenty-seven percent (*n* = 23) of students belonged to multiple URGs. Among the students in URGs, three out of four experienced distress. The students in URGs experienced distress at a higher rate compared to their peers in all categories. Twenty percent (*n* = 16) of students did not agree that there was equitable treatment on campus. This rate was higher among students in URGs compared to their peers. Eighty-three percent (*n* = 52) of individuals who did not participate in an affinity group recognized there was a need. Among individuals who did not participate, nearly half identified with a URG. Pharmacy students, especially those in URGs, may benefit from additional interventions by pharmacy schools to improve the offering and visibility of supportive services. Providing URG-centered resources addresses the gap between the wellbeing and academic experience of pharmacy students in URGs compared to their peers.

## 1. Introduction

Pharmacy students experience a higher rate of stress and poorer wellbeing compared to students in other health science graduate programs, a population that has already been flagged for a decline in mental wellbeing [1,2]. Pharmacy students have identified their learning environment, student engagement, social connectedness, and student support as being important factors that influence their wellbeing [3].

Typically, pharmacy students belonging to underrepresented groups (URGs) experience poor wellbeing at higher rates than their peers due to obstacles in their learning environment where they endure inadequate displays of cultural competency, lack of protection of vulnerable identities, and poorer student experience due to higher rates of microaggressions [4,5,6]. The consequences of high distress and poor wellbeing among pharmacy students in URGs include experiencing burnout, worse mental and physical health outcomes, and detachment from or distrust in their institution [7,8,9,10]. A novel strategy to combat these barriers and improve the experience of pharmacy students in URGs is through the support of affinity groups.

Affinity groups are defined as designated spaces tailored for individuals of specific identities centered around mentoring, improving access to resources, and providing a buffer from any stereotypes, social isolation, or imposter syndrome that the individual may experience [11]. Affinity groups serve as microenvironments of underrepresented groups in spaces where they may be otherwise isolated. Themes highlighted in affinity groups include improving a sense of belonging, interactions with classmates, and available or desired support [12]. These areas are congruent with rectifying the negative experiences of underrepresented individuals and may serve as protective measures.

College-supported affinity groups were formed in a Midwestern College of Pharmacy (COP) in 2019 after focus groups composed of students, staff, and faculty identified a need for increased support for underrepresented students. The groups were designed as identity-based social support groups available to all students, supported by university faculty and staff. They were structured to have at least one faculty or staff member lead the group and were accessible to students, staff, and faculty by signing up online or receiving an invitation to join by word of mouth. This created an environment where students were invited to make a conscious effort to participate and seek support. Each group set their own agendas on how they wanted to utilize the affinity space to support their needs.

The structure and success of affinity groups have seldom been studied in pharmacy, with most of the literature centering around their place in the workplace environment or in other higher education spaces [13]. This study aims to examine the current wellbeing among pharmacy students in URGs and investigate the impact on having access to affinity groups.

## 2. Methods

This study is a web-based cross-sectional survey administered anonymously to all pharmacy students at the college of pharmacy in April and May of 2023. The survey was centered around the concepts of diversity, equity, inclusion, and accessibility (DEIA) efforts at the COP and their impact on student wellbeing and student experiences with the college-supported affinity groups.

The survey design was based on past quality improvement surveys authored by current students and questions asked in the Student Experience in the Research University Survey (gradSERU) survey, a survey centered around student experience conducted by the University (Appendix A) [14]. The incorporation of past survey designs served as both comparators to the current student population and as a validation tool. The survey was sent via email to all students currently enrolled in the PharmD program during spring semester 2023 (n = 477). The survey did not contain any identifiable information and responses were encrypted to require a university log in to complete.

The 22-question survey was broken into four main components: demographic information, student distress, and affinity group involvement (Table 1) (Appendix A, Appendix B and Appendix C).

The demographic section included a question aimed to identify individuals in URGs as defined by federal protected classes: race/ethnicity, sexual orientation, religion, disability, family status, and nationality [15]. First-generation students, individuals who are the first person in their family to go into higher education, were included as an URG due to their well-documented challenges in attending pharmacy school and other iterations of higher education [16].

The student wellbeing section included one dichotomous-response-styled question related to any distress in 5 different areas experienced in the two weeks previous to completing the survey and 15 5-point Likert-scale questions, ranging from strongly disagree to strongly agree, about influencing factors from the COP related to student support and wellbeing (Appendix A and Appendix B).

The third section asked questions based on the students’ participation in affinity groups. These questions centered around the effectiveness of affinity groups, barriers to participation, and ways that these groups may be improved upon (Appendix C).

A singular investigator reviewed the de-identified survey data. Descriptive statistics, both frequencies and proportions, were used to analyze the data based on the study aims. All submissions (n = 86) were included in the final data. There were no submissions that met the exclusion criteria of missing data or straight-line answers, answering all of the questions in the same manner. Subgroup analysis was conducted to identify differences among different subpopulations including students who identified as a URG compared to those who did not, and students who participated in affinity groups compared to those who did not participate. This study was approved by the IRB committee of the supporting institution.

## 3. Results

### 3.1. Demographics

A total of 86 pharmacy students completed the survey, reflecting a response rate of 18% among pharmacy students (Table 1). Ninety-four percent (n = 81) of students were in their first or second year of pharmacy school, accounting for an average of 35% between the two classes. Sixty-five students (75%) identified with at least one URG. Among the identities represented, first-generation students (n = 36), underrepresented racial/ethnic groups (n = 24), and LGBTQ+ (n = 13) were the three most prominent groups. Twenty-seven percent (n = 23) of students belonged to two or more URGs.

### 3.2. Current Distress

Seventy-nine percent (n = 68) of all students surveyed reported experiencing distress of some kind within the last 2 weeks (Table 1). Nine out of ten of the reported distressing experiences were related to mental health. This was followed by distress related to emotional wellbeing, which accounted for over half of all students reporting distress. Distress related to physical health, safety and security at school, and safety and security at home were all reported at a lower rate. Among URG students, three out of four students reported experiencing distress. Students in URGs experienced distress at a higher rate compared to their peers in all categories. Seventy-three percent (n = 36) of students in URGs reported experiencing multiple forms of distress.

### 3.3. Availability of Resources

One-third of all students did not agree with the provision of adequate resources by the COP to address wellbeing (Table 1). This is further highlighted by students’ answers to their knowledge of resources where, among the five resources described, an average of 66% (n = 57) of students agreed that they knew about each resource. Students were the least knowledgeable about affinity groups and emergency financial aid resources.

### 3.4. Sense of Belonging

One in five students reported that they did not feel as though they belonged as a COP student (Table 1). This rate was similar regardless of URG status. Examining student’s perception of community, 87% (n = 75) of students reported having a friend within their cohort. Students in URGs reported a higher rate of agreeing with the statement classmates cared about their wellbeing compared to their peers.

Twenty percent (n = 16) of students did not agree that there was equitable treatment on campus (Table 1). This rate was higher among students in URGs compared to their peers. This sentiment is observed in students’ beliefs about the climate that students in URGs experience. Overall, 37% (n = 32) of students agreed that students in URGs experience an inequitable climate. Among students in URGs, 40% (n = 25) held this belief, a higher percentage compared to their peers.

### 3.5. Perception of Academic Experience

While one-third of students did not agree that the COP cared about their academic progress, this rate was seen at an even higher rate among students in URGs. This is further supported by the difference in how students experienced communication with COP faculty, a discrepancy of 25% between students in URGs and their peers. Overall, nearly 40% (n = 52) of students did not agree that there was open communication between students and faculty (Table 1). Conversely, a higher percentage of students in URGs felt that faculty were available to talk with them, an indication that access to faculty does not necessarily correlate with connection and understanding.

### 3.6. Affinity Group Participation

One in four students belonged to an affinity group, all of whom identified with at least one URG (Table 2). Among these individuals, only eight (34%) students believed their current structure to be effective. Ninety-one percent (n = 21) of affinity group participants identified at least one barrier to their activity level in the group. The most common barriers included poor attendance from other students and faculty, poor communication, and conflicting scheduling. Suggestions for improvement in affinity group structure included in-person meetings, expanding to include other health science graduate programs, improved communication, and increased advertisement.

Eighty-three percent (n = 52) of individuals who did not participate in an affinity group recognized there was a need (Table 2). Among individuals who did not participate, nearly half identified with a URG. One reason students noted for not participating was because of the lack of representation for the URG that they identified with. Suggestions for support of identities not currently represented in the COP offered affinity groups included Muslim, Caribbean/West Indian, little people, and caregivers. One student noted that another obstacle to their desire to participate in an affinity group was that they did not feel safe or respected by the COP.

Examining differences in answers centered around perceived support and sense of belonging between those who did and did not participate in affinity groups, those in affinity groups had a higher rate of agreement related to cohort community including reporting having friends in their cohort and believing that their classmates cared about their wellbeing. Questions centered around relationships with faculty, communication, and equitable treatment favored those who were not in affinity groups.

## 4. Discussion

Perceived support was the main finding that emerged from the survey results favoring a more positive experience among affinity group participants compared to non-participants, aligning with previous literature relating to student engagement and success. Students who participated in affinity groups showed higher rates of engagement with their classmates and perception of support from their peers compared to those who did not.

### 4.1. Impact of Support on Student Engagement

It is well documented that social support in the form of positive student–peer relationships, student–faculty relationships, and the communication of a sense of purpose to students positively impacts their engagement in the academic experience and improves their perception of their academic workload and performance [17]. While student–peer relationships were positively impacted among affinity group participants, the perception of the academic experience was worse compared to students who did not participate in affinity groups. This could be due, in part, to the lack of representation of individuals from URGs in faculty and staff positions despite their presence in social group settings such as affinity groups. The American Association of Clinical Pharmacy [18] reported that only 4.7% and 2.9% of pharmacy faculty were African American and Hispanic, respectively. This faculty representation is even smaller among other URGs and remains a major hindrance for pharmacy schools’ recruitment of students in URGs [19]. Lastly, examining communication, students in URGs experienced dissonance with faculty twice as often as their peers. Without open communication, there is a lower likelihood of communicating purpose, resulting in a worse sense of belonging among students in URGs [20].

Despite this lag in URG visibility in academic spaces, addressing non-academic measures for support is key to improving student accomplishments. It has been documented that perceived emotional support and prioritizing the student’s wellbeing has a larger impact on student achievement compared to academic support [21]. Addressing social support in the academic setting has also led to an improvement in life satisfaction and academic motivation, as well as a reduction in burnout, all of which positively contribute to academic engagement and positively impact the wellbeing of students [22,23].

### 4.2. Impact of Support on Student Wellbeing

Despite the positive benefits that some individuals have experienced through affinity groups, there remains uncertainty regarding how they have impacted wellbeing. Social support is indicated in the literature for positively influencing the wellbeing of both students and underrepresented individuals [24,25]. The research has focused primarily on measures of social, emotional, and academic health, with major themes of mental health and quality of life highlighted throughout. The dissection of the results commonly addresses the importance of the accessibility of various networks and communities [26]. The availability of resources is another area of importance in relation to student wellbeing [27]. One-third of students were not knowledgeable about resources offered through their pharmacy program that are designed to have a positive impact on wellbeing. Connecting this gap between availability and access to resources is important for improving student wellbeing.

### 4.3. Barriers to Affinity Group Engagement

Among affinity group participants, the top barriers to participation included poor attendance of other members, conflicting scheduling, and poor communication. An explanation for poor attendance resides in the theory that there is a burden on facilitators and group members to promote and sustain affinity groups. Past research has explored the added burden that academic faculty as well as community members who identify with URGs have in creating space for themselves and introducing DEIA policy into settings that systemically exclude these groups [28]. Conflicting scheduling, or, from another perspective, the lack of prioritization of affinity group participation, becomes a more complicated issue when considering the different levels of urgency in competing priorities and if the benefits received from participation in affinity groups outweighs the need to prioritize other tasks and responsibilities. A study examining pharmacy student motivation in community-engaged learning participation found that competing priorities, time organization, and not feeling like they were contributing all played a role in barriers to engagement = [29]. Perceived low prioritization could also be a result of the dynamic membership due to student turnover. Lastly, poor communication and implied low motivation to engage can impact an individual’s feelings of connection to a group [30]. This could be due to the added burden of stigma in associating with an affinity group. Individuals may experience vulnerability in highlighting parts of their identity from a URG [31]. Having this added burden to participation could negatively impact an individual’s ability and willingness to participate.

### 4.4. Improving Affinity Group Utilization

The barriers described are multifaceted but can be linked to the need for prioritization of affinity groups in order to improve engagement. Innovating affinity groups to make them a larger priority can be achieved by an individualized and dynamic design.

Creating a more individualized experience for potential affinity group participants may help increase the engagement with and utility of the group. Contrarily, as the point of an affinity group is to provide a certain level of structured unity, it may be beneficial to introduce separate programming centered around one-on-one relationships among individuals from URGs such as concordant mentoring by faculty in affinity groups where students have the opportunity to receive tailored interventions that are more impactful to their needs. This version of intentional and culturally responsive mentorship has been demonstrated to be of value to students who identified the importance of having someone with their same identity in a mentorship role both in gender identity and race/ethnicity [32,33]. Having faculty more notably engage with students in affinity groups may help improve the perception of faculty approachability and communication among students from URGs.

In addition to individualizing the experience of students, creating a more flexible structure to affinity groups and increasing the frequency of meeting opportunities may improve their utility. While the current group design allows members of the different affinity groups to choose their preferred method of communication, frequency of meeting, and manner in which they support one another, this becomes less ideal if there are differences within the group of how students desire support. As it currently stands, many students reported that if their group met over the lunch hour, they were often prohibited from attending due to competing commitments with other student organizations or leadership opportunities that students participate in.

### 4.5. Limitations

The main limitation to this work is the distribution of participation among the different pharmacy cohorts and low overall participation among pharmacy students, potentially impacting the generalizability of findings. Participation among first- and second-year students made up 94% of the responses. This could contribute to the underpowering of the study due to the small sample size among different subgroups. This is partly due to their presence on campus and the ability to directly advertise the survey to their cohorts compared to third- and fourth-year students who only received advertisement of the survey through email due to their remote option access for class attendance and completing off-campus clinical rotations. This could impact an individual’s level of participation in the affinity group due to differences in accessibility and resources. Addressing overall participation, it was difficult to assess how many students were categorized as belonging to URGs due to the lack of reporting of this information. It is assumed that the overall number of students identifying with URGs is lower than the 75.5% (n = 65) that made up participants, an indication of overrepresentation of this viewpoint. The identities collected by the COP include racial and ethnic minorities—27.9% (n = 24) of survey participants compared to 33.7% (n = 161) of students enrolled—and first-generation students, who accounted for 41.9% (n = 36) of survey participants compared to 28.3% (n = 135) of students enrolled.

Another limitation is that not all URGs are represented in the current offering of affinity groups and represent a very heterogeneous perspective of identity. Since this survey was administered, two additional affinity groups have been added: the Muslim and caregiver affinity groups.

In addition to this, deeper assessment was lacking in the realm of exploring support and other contributing factors to wellbeing outside of the academic environment, all of which may have a great influence on student wellbeing. Future research should continue to identify approaches to improve student wellbeing. Another component of limitations to the data includes the lack of comparison between the demographics of the study participants compared to the student population due to the lack of identity information collected by the COP.

## 5. Conclusions

Pharmacy students, especially those in URGs, may benefit from additional interventions by pharmacy schools to improve the offering and visibility of supportive services. Education highlighting physical and mental health, wellbeing and safety, and dedicated support and resources in these areas should be prioritized given the high incidences of distress. Providing URG-centered resources may help address the gap between wellbeing and the academic experience of pharmacy students in URGs compared to their peers. Affinity groups provide a blueprint for providing social support in an academic environment and, if properly implemented, may help improve wellbeing and engagement among students in URGs.

## Figures and Tables

**Table 1 pharmacy-13-00070-t001:** Student demographic characteristics (n = 86).

Characteristics	Students (%)
Students by Graduation Year	86 (100)
2023	3 (3.5)
2024	1 (1.2)
2025	30 (34.9)
2026	52 (60.5)
Underrepresented Group (URG)	65 (75.5)
Racial/ethnic minority	24 (27.9)
Religious minority	3 (3.4)
LGBTQ+	13 (15.1)
Veteran Status	1 (1.2)
First General College Student	36 (41.9)
Person with disabilities	8 (9.3)
International student	6 (7.0)
Parent	1 (1.2)
Two or more URGs	23 (26.7)
No URG	21 (24.4)
Recent distress (Y/N)	68 (79.0)
Physical health	19 (27.9)
Mental health	62 (91.2)
Emotional Wellbeing	49 (72.1)
Safety/security at school	6 (8.8)
Safety/security at home	1 (1.5)
College of Pharmacy (COP) Support *	
Adequate resources for wellbeing	57 (66.3)
My classmates care about my wellbeing	65 (75.6)
COP cares about my academic progress	58 (67.4)
Friendship within cohort	75 (87.2)
Faculty member accessibility	62 (72.1)
Campus Experience *	
Equitable treatment on campus	70 (81.4)
Climate is equal for URGs	54 (62.8)
Encouragement of diversity expression from faculty	62 (72.1)
Communication between faculty and students	54 (62.8)
Belonging as COP student	69 (80.2)
Resource Knowledge *	
Physical health	62 (72.1)
Mental mealth (University staff)	70 (81.4)
Mental health (COP staff)	60 (69.8)
Emergency financial aid	36 (41.9)
Affinity groups	54 (62.8)

* = 5-point Likert scale portion of survey. Score based on individuals who agreed with the sentiment (answered ‘Strongly Agree’ or ‘Agree’). Summarized version of survey questions (Appendix A).

**Table 2 pharmacy-13-00070-t002:** Affinity group (AG) participation (n = 86).

	Students (%)
Affinity Group Participants Effectiveness of affinity groups Involvement in AG outside of COP Barriers that impact level of activity in AG Poor peer/faculty attendance Poor communication Meeting times conflict with my schedule Content is not engaging I am the only one on my campus to attend the AG	23 (26.7) 8 (34.8) 7 (30.4) 21 (91.3) 9 (42.9) 9 (42.9) 10 (47.6) 5 (23.8) 1 (4.8)
Non-Affinity Group Participants There is a need for affinity groups * I identify with an identity represented in the currently available affinity groups	63 (73.3) 52 (82.5) 30 (47.6)

* = 5-point Likert scale portion of survey. Score based on individuals who agreed with the sentiment (answered ‘Strongly Agree’ or ‘Agree’).

## Data Availability

Data is unavailable due to privacy restrictions.

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
