# Peer review of "Affinity Group Experiences in Pharmacy Education: A Pilot Study"

_pharmacy, 2025, doi:10.3390/pharmacy13030070_

Round 1
Reviewer 1 Report
Comments and Suggestions for Authors
Thank you for the opportunity to review your manuscript, “Affinity Group Engagement in Pharmacy Education: A Pilot Study.” While the topic is timely and addresses an important gap in pharmacy education related to student well-being and support for underrepresented groups (URG), several major issues with the study design, conceptual framing, and findings interpretation must be addressed prior to publication.
Comment 1
The primary concern is the small and potentially non-representative sample. The response rate was only 18% (n=86 out of 477 students), and the sample was skewed toward first- and second-year students (94% of respondents). This is concerning in terms of sampling bias and limits the generalizability of results. Equal division across all cohorts is necessary to confirm the conclusions drawn.
Comment 2
There is no operational definition of “affinity group” for the purposes of this study provided in the paper. It is not stated whether the groups were formally institutionalized, student-run, identity-based, or simply social support groups. This affects the interpretability of the findings, especially because many students were clearly unaware of or uninvolved with the groups.
Comment 3
The URG grouping represents a very heterogeneous set of identities (e.g., racial/ethnic minorities, LGBTQ+ students, first-generation college students, parents, students with disabilities, international students, etc.). While such inclusivity is undoubtedly well-intentioned, it detracts from the specificity of the findings and may obscure distinct experiences of marginalization.
Comment 4
The current title fails to capture the main objective or thrust of the study. The focus is not simply "engagement" but also perceived well-being, support, barriers to involvement, and overall campus climate.
Comment 5
While descriptive statistics are provided, no inferential statistics are used to establish group differences (e.g., URG vs. non-URG, affinity group members vs. non-members). Without this, group comparisons are speculative in nature regarding significance.
Comment 6
The authors state that “framework analysis” was used to identify themes related to study aims. The manuscript does not include, however, an explanation of what qualitative data were collected and how framework analysis was conducted in this research. Framework analysis is typically used for the analysis of open-ended qualitative data such as interview transcripts, focus groups, or responses to open-text survey questions. In this research, the vast majority of data are quantitative (e.g., Likert scales, multiple choice questions), and there is no visible presentation of qualitative data (e.g., thematic quotes or narrative responses). If open-ended responses were present and analyzed qualitatively, the authors must (1) specify where those data are shown in the survey, (2) explain the analytical process (e.g., coding strategy, number of coders, matrix construction), and (3) present illustrative quotes in the results or discussion to support themes. In the absence of these elements, it is unclear how framework analysis was applied in a meaningful way and how the “themes” presented emerged from the data.
Author Response
Thank you for taking the time to review this! I have included replies to your comments linked below.
- Comment 1: The primary concern is the small and potentially non-representative sample. The response rate was only 18% (n=86 out of 477 students), and the sample was skewed toward first- and second-year students (94% of respondents). This is concerning in terms of sampling bias and limits the generalizability of results. Equal division across all cohorts is necessary to confirm the conclusions drawn.
- I attempt to address this in the limitations and have added language to support this point and the potential concern for generalizability. “The main limitation to this work is the distribution of participation among the different pharmacy cohorts and low overall participation among pharmacy students, potentially impacting the generalizability of findings.” and “Addressing overall participation, it was difficult to assess how many students were categorized as URG due to the lack of reporting of this information. It is assumed that the overall number of students identifying with URG is lower than the 75.5% (n = 65) that made up participants, an indication of overrepresentation of this viewpoint. The identities collected by the COP include racial and ethnic minorities, 27.9% (n = 24) of survey participants and 33.7% (n = 161) of students enrolled, and first generation students, 41.9% (n = 36) of survey participants compared to 28.3% (n = 135) of students enrolled.”
- Comment 2: There is no operational definition of “affinity group” for the purposes of this study provided in the paper. It is not stated whether the groups were formally institutionalized, student-run, identity-based, or simply social support groups. This affects the interpretability of the findings, especially because many students were clearly unaware of or uninvolved with the groups.
- The third paragraph of the introduction section (line 52) covers how affinity groups are defined in previous literature. In addition to this, I inserted language to explicitly say that our intention was for their structure and how that manifested in the design. “College supported affinity groups were formed in a Midwestern College of Pharmacy (COP) in 2019 after focus groups composed of students, staff, and faculty identified a need for increased support for underrepresented students. The groups were designed as identity-based social support groups available to all students, supported by university faculty and staff. They were structured to have at least one faculty or staff member lead the group and were accessible to students, staff, and faculty by signing up online or receiving an invitation to join by word of mouth.
- Comment 3: The URG grouping represents a very heterogeneous set of identities (e.g., racial/ethnic minorities, LGBTQ+ students, first-generation college students, parents, students with disabilities, international students, etc.). While such inclusivity is undoubtedly well-intentioned, it detracts from the specificity of the findings and may obscure distinct experiences of marginalization.
- I attempted to address this in the limitations by saying that we are expanding the groups and inserted more specific language to show that this is a flaw we are eventually hoping to rectify, “Another limitation is that not all URG are represented in the current offering of affinity groups and represent a very heterogeneous perspective of identity.”
- Comment 4: The current title fails to capture the main objective or thrust of the study. The focus is not simply "engagement" but also perceived well-being, support, barriers to involvement, and overall campus climate.
- Change Affinity Group Engagement in Pharmacy Education: A Pilot Study → “Affinity Group Experiences in Pharmacy Education: A Pilot Study”
- Comment 5: While descriptive statistics are provided, no inferential statistics are used to establish group differences (e.g., URG vs. non-URG, affinity group members vs. non-members). Without this, group comparisons are speculative in nature regarding significance.
- The data presented in this report is intentionally experience-driven and grounded in qualitative insights, rather than structured for inferential statistical comparisons. Our goal was to elevate authentic narratives from participants, particularly those from historically excluded backgrounds, to better understand their lived experiences within our program context. Subset analyses that attempt to quantify differences (e.g., URG vs. non-URG, affinity group members vs. non-members) risk oversimplifying or distorting the richness and complexity of these narratives. Given the qualitative nature of the data, we believe that such statistical comparisons would not provide a meaningful or representative reflection of the participants’ voices or the emergent themes.
- Comment 6: The authors state that “framework analysis” was used to identify themes related to study aims. The manuscript does not include, however, an explanation of what qualitative data were collected and how framework analysis was conducted in this research. Framework analysis is typically used for the analysis of open-ended qualitative data such as interview transcripts, focus groups, or responses to open-text survey questions. In this research, the vast majority of data are quantitative (e.g., Likert scales, multiple choice questions), and there is no visible presentation of qualitative data (e.g., thematic quotes or narrative responses). If open-ended responses were present and analyzed qualitatively, the authors must (1) specify where those data are shown in the survey, (2) explain the analytical process (e.g., coding strategy, number of coders, matrix construction), and (3) present illustrative quotes in the results or discussion to support themes. In the absence of these elements, it is unclear how framework analysis was applied in a meaningful way and how the “themes” presented emerged from the data.
- Removed framework analysis from the description of how themes were identified. Replace with “A singular investigator reviewed the de-identified survey data. Descriptive statistics, both frequencies and proportions, were used to analyze the data based on the study aims.”

Reviewer 2 Report
Comments and Suggestions for Authors
Dear Authors,
This is very usefull study dealing with necessity for URG supportive services.
Introduction: is well structured
Methodology:
Have there be prior study calculation what is the number of students that should be involved in order to have statistically significant conclusions?
If survey is web based how it was determined that only pharmacy students filled survey? It was sent to all students via mail?
What statistical analysis were used?
Was survey authorised ant its questions? Is it based on questions in similar studies?
Results:
All percentages are based on 18% of population, have you calculated what would be differences if more student population have had participated?
Limitation of the study should also be that low number of students participated in the study and might be that mostly students in URG participated.
Conclusion should have more detailed information which supportive services should be offered.
Best of luck
Author Response
Methodology:
- Have there be prior study calculation what is the number of students that should be involved in order to have statistically significant conclusions?
- This project was designed to capture experience-based data, and as such, statistical significance calculations for sample size were not a component of the study design. The evaluation reflects voluntary participation from students who accepted the invitation to provide feedback on their experiences within the College of Pharmacy. Because the intent was to elevate qualitative insights and subjective reflections rather than to generalize findings across a population, traditional statistical thresholds for significance were not applied. Our focus remains on understanding and honoring the voices of those who chose to engage.
- If survey is web based how it was determined that only pharmacy students filled survey? It was sent to all students via mail?
- Text reflects that only pharmacy students received an email to fill out the survey and promotion during their scheduled lecture time. In addition to this, the text refers to the safeguard that only students with a university specific email address were able to complete the survey. Text was updated to show that it was sent via email. “The survey was sent via email to all students currently enrolled in the PharmD program during spring semester 2023 (n = 477). The survey did not contain any identifiable information and responses were encrypted to require a university log in to complete.”
- What statistical analysis were used?
- Descriptive statistics was used to assess the Likert scales and experiences of current distress among affinity group and non-affinity group participants. “A singular investigator reviewed the de-identified survey data. Descriptive statistics, both frequencies and proportions, were used to analyze the data based on the study aims.”
- Was survey authorised ant its questions? Is it based on questions in similar studies?
- Survey and included questions was authorized by the university IRB. The text refers to previous studies/surveys that include similar or congruent questions that served as inspiration for this study. “The survey design was based on past quality improvement surveys authored by current students and questions asked in the Student Experience in the Research University Survey (gradSERU) survey, a survey centered around student experience conducted by the University (Appendix B) (Graduate School, 2023)”
- Results:
- All percentages are based on 18% of population, have you calculated what would be differences if more student population have had participated?
- No. In the limitations we caution that the low overall participation rate among pharmacy students may impact the generalizability and compare two of the identity categories to information collected by the university to show the difference between the sample size and the overall population, as outlined below.
- Limitation of the study should also be that low number of students participated in the study and might be that mostly students in URG participated.
- Text is updated to reflect this and information from college collected stats on student demographics provided as a comparison. “Addressing overall participation, it was difficult to assess how many students were categorized as URG due to the lack of reporting of this information. It is assumed that the overall number of students identifying with URG is lower than the 75.5% (n = 65) that made up participants, an indication of overrepresentation of this viewpoint. The identities collected by the COP include racial and ethnic minorities, 27.9% (n = 24) of survey participants and 33.7% (n = 161) of students enrolled, and first generation students, 41.9% (n = 36) of survey participants compared to 28.3% (n = 135) of students enrolled.”
- Conclusion should have more detailed information which supportive services should be offered.
Language was added to support this. “Education highlighting physical and mental health, wellbeing and safety as well as dedicated support and resources in these areas should be prioritized given the high incidences of distress”.
Round 2
Reviewer 2 Report
Comments and Suggestions for Authors
Dear Authors,
Thank you for the reply. There is noted limitations, however it is an interesting paper.
Best of luck